# Socio-demographic and neighbourhood factors influencing urban green space use and development at home: A population-based survey in Accra, Ghana

A. Kofi Amegah[1]*, Kelvin Yeboah[1], Victor Owusu[2], Lucy Afriyie[2], Elvis Kyere-Gyeabour[3], Desmond C. Appiah[4], Patrick Osei-Kufuor[5], Samuel K. Annim[2,6], Samuel Agyei-Mensah[3], Pierpaolo Mudu[7]

1 Department of Biomedical Sciences, Public Health Research Group, University of Cape Coast, Cape Coast, Ghana, 2 Ghana Statistical Service, Finance Close, Accra, Ghana, 3 Department of Geography and Resource Development, University of Ghana, Legon, Accra, Ghana, 4 Clean Air Fund, Ghana Country Office, Accra, Ghana, 5 Department of Peace Studies, School for Development Studies, University of Cape Coast, Cape Coast, Ghana, 6 Department of Applied Economics, School of Economics, University of Cape Coast, Cape Coast, Ghana, 7 World Health Organization, Geneva, Switzerland

* aamegah@ucc.edu.gh

**Data Availability Statement:** All relevant data are within the manuscript and its Supporting Information files.

## Abstract

In Sub-Saharan Africa and other developing regions, there has been very little systematic attempt to document the uses and perceived health benefits of urban green spaces in cities and the factors influencing usage. We therefore sought to establish the availability, accessibility and use of urban green spaces, and the perceived health benefits in an African population. We also ascertained the factors influencing use and development of green spaces at home. A population-based survey was conducted in Accra, the capital city of Ghana, spanning 11 Municipal and 3 Sub-Metropolitan areas. Multivariable binary logistic regression adjusting for potential confounders was used to establish the association between green space use and development at home, and socio-demographic, neighbourhood and health factors. Odds ratios and their corresponding 95% confidence intervals were estimated from the models. Several socio-demographic (gender, age, marital status, occupation, ethnicity, religion) and district-level (population density, income level, neighbourhood greenness) factors were associated with use of green spaces and development of green spaces at home in Accra. Residents who were worried about depletion of green spaces in their community were more likely to develop green spaces at home. In neighbourhoods with moderate and high level of greenness, residents were less likely to develop green spaces at home. Five-percent and 47% of green space users in Accra reported witnessing an improvement in their physical and mental health, respectively, from use of green spaces. The study findings can inform policy action for promoting use and development of green spaces in African cities and for mitigating depletion and degradation of the limited urban greenery.

**Funding:** The field data collection was made possible by a consultancy grant from the World Health Organization under the Accra Urban Health Initiative (WHO Unit Reference: AQH/ECH; Purchase Order: 202755480). The funder had no role in the study design; in the collection, analysis, and interpretation of data; in the writing of the report; or in the decision to submit the article for publication.

**Competing interests:** The authors have declared that no competing interests exist.

## Introduction

Urban green spaces such as gardens, zoos, parks, and suburban natural areas and forests, have increasingly gained global attention owing to their role in promoting good health within populations [1–3]. They serve recreational and aesthetic purposes thus providing the needed suitable ecosystem services in most cities [4–6]. There is increasing evidence from a number of empirical studies and reviews on the role of green spaces in improving health and well-being especially cardiovascular and mental health [2,3,7]. The health benefits of green spaces occur through pathways such as ameliorating the effect of environmental exposures such as temperature, air pollution and noise, encouraging healthy behavioural practices such as physical activity, and fostering social cohesion [2,7].

In spite of the benefits conferred by urban green spaces, growing evidence suggest the depletion of urban greenery in many cities globally [8]. In a study of 386 European cities, a dramatic drop in per capita green space provision in cities with high population density was noted [9]. The authors also found access to greenspace to rapidly decline as cities grow. Many Eastern European cities have experienced a decline in greenspace following post-socialist changes [10,11]. Similarly, McDonald et al. [12] found an open space loss between 1990 and 2000 for all 274 metropolitan areas in the contiguous United States that they examined in their study. In China, mixed results have been noted with some cities witnessing a decline in green spaces and others, an increase [13]. Cities in many developing countries are, however, losing green spaces at a rapid pace [13]. Whilst significant strides are being made in most western countries to reverse this depletion, in most African countries, the situation continues to worsen [1,14,15]. The threat to urban green spaces is mainly driven by the increasing rate of urbanization in countries. As a result of rapid urbanization, green spaces are increasingly being developed into residential settlements. About half of the world's population live in urban areas with projections estimating the urban population to increase to 68% by 2050 [16]. The fastest urbanizing regions in the world are Sub-Saharan Africa and South Asia [16] and accounts for the fast decline in urban green spaces in these regions.

In Ghana, Accra and Kumasi have seen an increase in built-up areas with significant reduction in forest vegetation cover in the two cities [17]. The destruction of forest vegetation cover observed in Ghanaian cities is driven by failure of public policy and lack of citizens will to protect green spaces [18,19]. According to Abass et al. [18], even though many citizens may have good knowledge of the benefits of green spaces, it often does not translate into the desire to adopt practical steps to conserve these spaces. The author further states that, in urban and peri-urban areas of Ghana, the use of land for residential, commercial and other economic purposes often far outweigh the need to preserve urban green spaces or use available land to create green spaces. Also, in many Ghanaian cities, the endemic problem of poor urban planning often leads to inappropriate use of land and the creation of slums at places where hitherto are supposed to be designated as green spaces [20].

There are notable health benefits of urban green spaces with majority of the evidence emanating from developed countries. In sub-Saharan Africa and other developing regions, there has been very little systematic attempt to document the uses and perceived health benefits of urban green spaces in countries, and the determinants. Availability of local evidence is very important to policy action for curtailing the decline in the limited urban greenery whilst also increasing the number of green spaces. In Ghana, a few case studies in Kumasi [18,21–24] and Sekondi [25] and a survey in Accra [20] have been conducted on the topic with mixed findings. A narrative review focused on the Accra–Tema area [19] has also been published. The survey conducted in Accra [20] was narrow in scope as it was not a population-based study and focused on only two slum communities, Nima and Agbogbloshie. This study conducted

spatio-temporal analysis to estimate the proportion of green spaces in the two communities between 2000 and 2018 and established residents' perceptions on urban green space availability and use. However, the study did not attempt to establish the factors influencing use of green spaces as well as the health benefits of use in the two communities.

It is against this background that we conducted a population-based survey in the Accra Metropolitan Area to establish availability, accessibility and use of urban green spaces, the perceived health benefits whilst also attempting to establish the factors influencing use and development of green spaces at home. We selected Accra because it is one of the rapidly growing city in Africa with an increasing population and unending infrastructural development which threatens the preservation of green spaces. The findings of the survey will inform policy action for addressing the depletion and degradation of urban spaces in Accra and other metropolitan areas of Ghana, developing green spaces in Accra and other cities, and providing preliminary data for conducting a large-scale epidemiological study to investigate the influence of urban green spaces on cardiovascular and mental health in Ghana.

## Materials and methods

### Study design and location

A population-based survey was conducted in the Greater Accra Metropolitan Area spanning 11 municipal areas namely; Ablekuma Central, Ablekuma North, Ablekuma West, Korley Klottey, Ayawaso East, Ayawaso Central, Ayawaso West, Ayawaso North, Okaikoi North and La Dadekotopon, and the 3 sub-metropolitan areas of Accra Metropolitan Assembly namely; Ablekuma South, Ashiedu Keteke and Okaikoi South.

The population density of the districts was classified into medium (1000–5000 Persons per sq. Km), high (5001–10000 Persons per sq. Km), very high (10001–20000 Persons per sq. Km) and extremely high (> 20000 Persons per sq. Km) according to the classifications proposed by United Nations Population Division and United Nations Human Settlements Programme (UN-Habitat). The districts were classified into low-, middle- and high-income based on the mean household income of the district as reported by the Ghana Living Standards Survey. Table 1 presents the population density and income levels of the districts.

**Table 1. Population density and income levels of the districts.**

| Districts | Population density | Income level |
|---|---|---|
| **Municipal Areas** | | |
| Ablekuma Central | Very high | Middle |
| Ablekuma North | Very high | Middle |
| Ablekuma West | Very high | Middle |
| Korley Klottey | High | Middle |
| Ayawaso East | Very high | Low |
| Ayawaso Central | Very high | Middle |
| Ayawaso West | Medium | High |
| Ayawaso North | Extremely high | Low |
| Okaikoi North | High | Middle |
| La-Dadekotopon | Medium | High |
| **Sub-Metropolitan Areas** | | |
| Ablekuma South | Very high | Low |
| Ashiedu Keteke | Extremely high | Low |
| Okaikoi South | High | Middle |

### Study population, sampling and data collection

A multi-stage sampling procedure was used to sample the study participants using the 2021 Ghana Population and Housing Census list of households. The first stage sampling units were the enumeration areas (EA) in the selected districts with a total of 200 EAs selected, known as the primary sampling unit (PSUs). The number of EAs selected in each district for inclusion in the sample was based on probability proportional to size (PPS). At the second stage, 2000 households were selected using systematic random sampling to form the secondary sample unit (SSU), allocated proportionately to the number of households in each district. The Supporting Information file provides information on the sampling strategy applied (S1 Table in S1 File). In each selected household, the household head and another adult member of the household where available were interviewed using a structured questionnaire administered through the CSPro platform with tablets. Fig 1 is a map of Greater Accra Metropolitan Area (GAMA) with an overlay of the households sampled.

Data collection took place in January, 2022. We collected information on sociodemographic characteristics of household heads; availability, state and use of green spaces; benefits of green spaces; and development of green spaces at home and changing patterns of green spaces in the neighbourhood/community.

### Assessment of improvements in mental health

We assessed improvements in mental health from use of green spaces as follows:

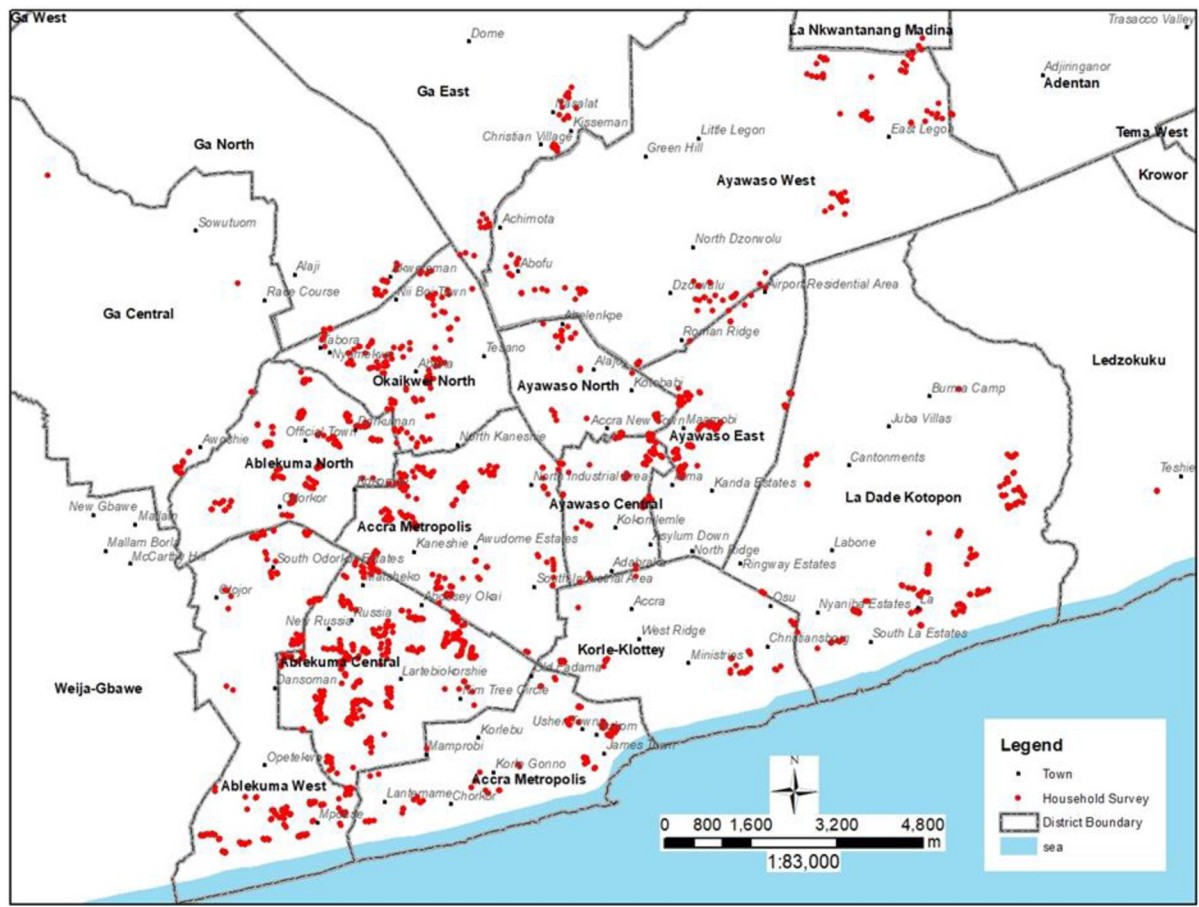

**Fig 1. Map of Greater Accra Metropolitan Area (GAMA) with an overlay of the households sampled.**

- For participants who indicated using green spaces, they were asked to indicate on a scale of (a) 1 (less) to 5 (more), whether they were more satisfied with themselves since they started using green spaces; (b) 1 (lower) to 5 (higher), whether they have noticed enhancement in their well-being (i.e. state of happiness); (c) 1 (no decrease) to 5 (high decrease), whether they feel a decrease in their stress levels; and (d) 1 (less) to 5 (more), they have developed more social connections while using green spaces.

- Principal component analysis was used to generate an index of the state of mental health

- The scores were split into four parts with quartile I, II, III and IV representing low, moderate, high and very high improvement, respectively.

## Generation of Normalized Difference Vegetation Index (NDVI) maps

NDVI maps for the years 2000, 2015 and 2020 were generated from bands in some satellite products. NDVI measures greenness and density of the vegetation captured in a satellite image and is calculated from the difference between two bands in the satellite products–visible red light band and near-infrared red (NIR) band. NDVI is the difference expressed as a number ranging from -1 to 1. The satellite images used were captured on the following dates: 2000– 2000/02/12 from Landsat 7, 2015–2015/01/01 from Landsat 8, and 2020–2020/08/13 from Landsat 8. Landsat 7 and 8 satellite images have a 30-meter resolution. Landsat 7 has been in operation since 1999 with Landsat 8 launched in 2013. Landsat 7 and 8 have NIR and red bands which can be directly used to compute NDVI from the satellite imagery. For Landsat 7, the NIR range is band 4 and red range is band 3 whereas for Landsat 8, the NIR range is band 5 and red range is band 4. The formula for computing the NDVI is:

$$NDVI = \frac{NIR - Red}{NIR + Red}$$

Landsat 7 NDVI formula is:

$$NDVI = \frac{Band\ 4 - Band\ 3}{Band\ 4 + Band\ 3}$$

Landsat 8 NDVI formula is:

$$NDVI = \frac{Band\ 5 - Band\ 4}{Band\ 5 + Band\ 4}$$

## Ethical considerations

We obtained ethical approval for the survey from the University of Ghana Ethics Committee for Humanities (Reference No: ECH 257/21-22) with the approval given on December 9, 2021. An informed consent was obtained from all study participants before they were interviewed. Information on the informed consent form was read to the selected participants by the research assistants with those consenting to participate in the study made to sign or thumbprint the form.

## Statistical analysis

For this study we restricted the analysis to information collected from only the household heads. Multivariable binary logistic regression adjusting for potential confounders was used to establish the association between green space use and development at home, and the socio-demographic, neighbourhood and health factors. Odds ratios and their corresponding 95% confidence intervals were estimated from the models.

## Results

Information on the socio-demographic characteristics of household heads are presented in Table 2. The proportion of male and female household heads were about equal (50.4% males vs. 49.6% females). About half of the household heads were aged 30 to 49 years with about 3% aged less than 20 years and 13% aged 60 years and above. About 9% of the household heads had no formal education with 16% educated up to tertiary level. About 82% of the household heads were Christians with Muslims making up 16%. About 35% and 32% of the household heads were Akans and Ga/Dangme, respectively. More than half (57%) of the household heads were married with 27% reporting never being married. About 42% of the household heads worked in the sales and services industry. About 31% of the household heads were manual workers, 10% were professional/technical/managerial workers, and 11% were unemployed.

Information on the availability, state and use of green spaces are presented in Table 3. About 31% of the household heads indicated that green spaces were available in their community/neighbourhood with the majority of the green spaces available being parks (42%). Trees along streets and road corners were also identified by 18.8% of the household heads. About 84% of the household heads indicated that the identified green spaces have either lost or are losing their greenness. Of the 31% of household heads who mentioned the availability of green spaces in their community/neighbourhood, only 9.5% reported using these green spaces with 30% of these respondents using the green spaces occasionally i.e. mostly on public holidays. About 54% of household heads using green spaces reported having used these green spaces for over 5 years. About 47% of household heads who use green spaces mentioned exercise as the main reason for accessing these green spaces. About 23% and 32% of these respondents indicated that, their homes were less than 50m, and 400m or more away from their preferred green spaces, respectively. Majority of these respondents (91%) walk to access these green spaces. About 83% of these respondents, whenever they access these green spaces, usually stay for between 2 and 4 hours. Of the household heads who indicated not using green spaces in spite of their availability in their community/neighbourhood, the reasons ranged from having to travel long distances (11.2%), not being suitable for children to play (26.7%), not well equipped for relaxation and excitement (14.5%), not in good state including loss of greenness (8.7%), and spaces being very noisy (4.4%) among others.

Information on the perceived benefits of the use of green spaces are summarized in Table 4. Of the respondents who indicated using green spaces, about 60% described themselves as being highly active. Only 5% of these respondents reported seeing an improvement in their physical health from use of green spaces. Between 51% and 54% of these respondents, reported deriving more satisfaction, higher level of happiness, developing more social connections and witnessing a higher decline in their stress levels from use of green spaces. About 49% of these respondents reported witnessing a very high improvement in their mental health from use of green spaces.

Information on development of green spaces at home and changing patterns of green spaces in the neighbourhood/community are presented in Table 5. Of the household heads interviewed, 21.2% reported having green spaces in the house with trees, flowers and lawns being the predominant green spaces found in these homes. Over half (51.7%) of the household heads identified improved air quality as the major reason for developing green spaces in the home. About 21% and 20% of the household heads identified aesthetics and enhancement of physical health as the major reason, respectively. Majority (87%) of the household heads felt the green spaces developed in the home were achieving their intended benefits. About 34% of the household heads felt the green spaces in their neighbourhood/community were depleting with 51% of them worried about the depletion and only 0.9% feeling that the local authority or

**Table 2. Sociodemographic characteristics of household heads (N = 1914).**

| Characteristics | Household Heads n (%) |
|---|---|
| **Gender** | |
| Male | 921 (50.4) |
| Female | 907 (49.6) |
| Missing | |
| **Age group (years)** | |
| <20 | 60 (3.2) |
| 20–29 | 340 (17.8) |
| 30–39 | 476 (24.9) |
| 40–49 | 479 (25.0) |
| 50–59 | 318 (16.6) |
| ≥ 60 | 241 (12.6) |
| **Educational level** | |
| No education | 163 (8.5) |
| Primary | 189 (9.9) |
| JHS | 541 (28.3) |
| SHS/Secondary/Technical | 718 (37.5) |
| Tertiary | 303 (15.8) |
| **Religion** | |
| Christian | 1571 (82.1) |
| Muslim | 306 (16.0) |
| Traditional/Spiritualist | 7 (0.4) |
| No religion | 26 (1.4) |
| Other | 4 (0.2) |
| **Ethnicity** | |
| Akan | 665 (34.7) |
| Ga/Dangme | 606 (31.7) |
| Ewe | 287 (15.0) |
| Guan | 29 (1.5) |
| Mole-Dagbani | 170 (8.9) |
| Grusi | 58 (3.0) |
| Gruma | 38 (2.0) |
| Mande | 11 (0.6) |
| Other | 50 (2.6) |
| **Marital status** | |
| Never married | 516 (27.0) |
| Married | 1091 (57.0) |
| Living together | 71 (3.7) |
| Divorced/Separated | 137 (7.2) |
| Widowed | 99 (5.2) |
| **Occupation** | |
| Professional/Technical/Managerial | 198 (10.3) |
| Clerical | 33 (1.7) |
| Sales and services | 802 (41.9) |
| Skilled manual | 424 (22.2) |
| Unskilled manual | 160 (8.4) |
| Agriculture | 16 (0.8) |
| Unemployed | 210 (11.0) |
| Other | 71 (3.7) |

**Table 3. Availability, state and use of green spaces.**

| | |
|---|---|
| **Availability of green spaces in community/neigbourhood (n = 1914)** | |
| Yes | 600 (31.4) |
| No | 1314 (68.7) |
| **Type of green spaces in community/neighbourhood (n = 600)** | |
| Parks | 253 (42.2) |
| Community gardens | 4 (0.7) |
| Cemeteries | 15 (2.5) |
| Playing field | 34 (5.7) |
| Forest/Woodlands | 7 (1.2) |
| Trees along streets and road corners | 113 (18.8) |
| Others incl. golf courses, private gardens | 79 (13.2) |
| Combination | 95 (15.8) |
| **Green spaces lost/losing its greenness (n = 600)** | |
| Yes | 503 (83.8) |
| No | 97 (16.2) |
| **Use of green spaces in community/neighbourhood (n = 600)** | |
| Yes | 57 (9.5) |
| No | 543 (90.5) |
| **Frequency of green space use (n = 57)** | |
| Daily | 8 (14.0) |
| 1x/Week | 14 (24.6) |
| 2-3x/Week | 13 (22.8) |
| 4-5x/Week | 2 (3.5) |
| Once/Month | 2 (3.5) |
| Twice/Month | 1 (1.8) |
| Occasionally/Public Holidays | 17 (29.8) |
| **Duration of green spaces use (n = 57)** | |
| <12 months | 10 (17.5) |
| 12–35 months | 8 (14.0) |
| 36–60 months | 8 (14.0) |
| >60 months | 31 (54.4) |
| **Reason for accessing green spaces (n = 57)** | |
| Exercise | 27 (47.4) |
| Appreciate nature i.e. watch wildlife and greenery | 1 (1.8) |
| Relaxation / Ease stress | 4 (7.0) |
| Recreation | 1 (1.8) |
| Socialization | 1 (1.8) |
| Others | 4 (7.0) |
| Combination | 19 (33.3) |
| **Proximity of house to preferred green space (n = 57)** | |
| <50m | 13 (22.8) |
| 50 – 99m | 4 (7.0) |
| 100 – 199m | 10 (17.5) |
| 200 – 399m | 12 (21.1) |
| ≥400m | 18 (31.6) |
| **Mode of transportation to preferred green spaces (n = 57)** | |
| Motorized | 4 (7.0) |
| Cycling | 1 (1.8) |

(*Continued*)

**Table 3.** (Continued)

| | |
|---|---|
| Walking | 52 (91.2) |
| **Length of stay during visit to preferred green space (n = 57)** | |
| ≤1 hour | 7 (12.3) |
| 2–4 hours | 47 (82.5) |
| 4–8 hours | 3 (5.3) |
| **Reasons for not using green spaces (n = 543)** | |
| Having to travel long distance | 61 (11.2) |
| No toilet facilities | 6 (1.1) |
| No sitting area | 14 (2.6) |
| Not in good state incl. loss of greenness | 47 (8.7) |
| Not well equipped for relaxation and excitement | 79 (14.5) |
| Not safe and secure | 3 (0.6) |
| Limited in size | 1 (0.2) |
| Very noisy | 24 (4.4) |
| Not suitable for children to play | 145 (26.7) |
| Other reasons | 147 (27.1) |
| Missing | 16 (2.9) |

community leaders are taking steps to stem the depletion. Only 0.6% of the household heads feel green spaces in their neighbourhood/community are improving.

Table 6 presents odds ratios from binary logistic regression analysis regressing socio-demographic characteristics of household heads on use of green spaces. Female household heads had 6.9 times higher odds (OR = 6.92; 95% CI: 3.13, 15.26) of using green spaces compared to their male counterparts. Household heads aged 30–49 years and ≥ 50 years had 2.9 and 3.1 times higher odds of using green spaces, respectively, compared to household heads aged ≤ 30 years (OR = 2.94; 95% CI: 1.48, 5.84 and OR = 3.10; 95% CI: 1.36, 7.06, respectively). Ga/Dangme household heads had 0.29 times lower odds (OR = 0.29; 95% CI: 0.14, 0.64) of using green spaces compared to Akan household heads. Compared to Christian, traditionalist/spiritualist household heads and those belonging to no religious group, had 0.1 times (OR = 0.11; 95% CI: 0.02, 0.62) lower odds of using green spaces. Compared to household heads who have never been married, married or cohabiting household heads had 1.9 times (OR = 1.88; 95% CI: 1.01, 3.52) higher odds of using green spaces. Compared to household heads who were skilled and unskilled manual workers, professional/technical/managerial household heads had 0.4 times lower odds of using green spaces (OR = 0.36; 95% CI: 0.15, 0.90).

Table 7 presents odds ratios from multivariable binary logistic regression analysis regressing socio-demographic characteristics of household heads on development of green spaces at home. Female household heads had 1.2 times (OR = 1.24; 95% CI: 0.97, 1.57) higher odds of developing green spaces at home compared to male household heads. Household heads aged ≥ 50 years had 0.6 times (OR = 0.60; 95% CI: 0.43, 0.85) lower odds of developing green spaces at home compared to household heads aged ≤ 30 years. Compared to household heads with no formal or primary education; JHS, SHS/Secondary/Technical and tertiary educated household heads had lower odds of developing green spaces at home with the odds ratio ranging from 0.21 to 0.43. Ewe/Guan household heads had 0.6 times (OR = 0.59; 95% CI: 0.43, 0.82) lower odds of developing green spaces at home compared to Akan household heads. Muslim household heads had 1.5 times (OR = 1.54; 1.02, 2.32) higher odds of developing green spaces at home compared to Christian household heads. Compared to household heads who have never been married, household heads who were married/living together (OR = 0.90;

**Table 4. Benefits of green spaces to users (n = 57).**

| Physical activity levels | N (%) |
|---|---|
| Level I (Inactive) | 1 (1.8) |
| Level II (Slightly active) | 1 (1.8) |
| Level III (Moderately active) | 7 (12.3) |
| Level IV (Active) | 14 (24.6) |
| Level IV (Highly active) | 34 (59.7) |
| **Improvement in physical health from use of green spaces** | |
| Yes | 3 (5.3) |
| No | 54 (94.7) |
| **Satisfaction with oneself from use of green spaces** | |
| Level I (Less satisfied) | 2 (3.5) |
| Level II | 7 (12.3) |
| Level III | 11 (19.3) |
| Level IV | 8 (14.0) |
| Level V (More satisfied) | 29 (50.9) |
| **State of happiness** | |
| Level I (Lower) | 3 (5.3) |
| Level II | 3 (5.3) |
| Level III | 13 (22.8) |
| Level IV | 7 (12.3) |
| Level V (Higher) | 31 (54.4) |
| **Decrease in stress levels** | |
| Level I (No decrease) | 2 (3.5) |
| Level II | 3 (5.3) |
| Level III | 15 (26.3) |
| Level IV | 6 (10.5) |
| Level V (High decrease) | 31 (54.4) |
| **Development of social connections** | |
| Level I (Less) | 3 (5.3) |
| Level II | 5 (8.8) |
| Level III | 8 (14.0) |
| Level IV | 12 (21.1) |
| Level V (More) | 29 (50.9) |
| **Improvement in mental health** | |
| Low | 14 (24.6) |
| Moderate | 14 (24.6) |
| High | 1 (1.8) |
| Very high | 28 (49.1) |

95% CI: 0.68, 1.18) and divorced/separated/widowed (OR = 0.90; 95% CI: 0.60, 1.35) had lower odds of developing green spaces at home, albeit the associations were not statistically significant. Compared to household heads who were skilled or unskilled manual workers, household heads who were in any of the following occupations; professional, technical, managerial, clerical, sales and services as well as those unemployed were also had higher odds of developing green spaces at home. The associations were, however, not statistically significant.

Table 8 presents odds ratios from multivariable binary logistic regression analysis regressing characteristics of district/neighbourhood and changing patterns of green spaces in neighbourhood/community on development of green spaces at home. Household heads who felt

**Table 5. Development of green spaces at home and changing patterns of green spaces in the neighbourhood/community.**

| | |
|---|---|
| **Availability of green space in house (n = 1914)** | |
| Yes | 406 (21.2) |
| No | 1508 (78.8) |
| **Type of green spaces in house (n = 406)** | |
| Trees and flowers | 315 (77.8) |
| Lawns | 13 (3.2) |
| Trees, flowers and lawns | 61 (15.1) |
| Plantation and garden | 7 (1.7) |
| Trees, flowers, lawns, plantation and garden | 9 (2.2) |
| **Reasons for developing green space in house (n = 406)** | |
| Aesthetics | 85 (20.9) |
| Enhance mental health | 1 (0.3) |
| Enhance physical health | 81 (20.0) |
| Enhance social health | 5 (1.2) |
| Improve air quality | 210 (51.7) |
| Other | 20 (4.9) |
| None | 4 (1.0) |
| **Feeling of green space at home achieving its intended benefits (n = 406)** | |
| Yes | 352 (86.7) |
| No | 54 (13.3) |
| **Feeling of green spaces depleting in neighbourhood/community (n = 1914)** | |
| Yes | 643 (33.6) |
| No | 1271 (66.4) |
| **Worried about the depletion (n = 643)** | |
| Yes | 325 (50.5) |
| No | 318 (49.5) |
| **Local authority or community leaders taking steps to stem depletion (n = 643)** | |
| Yes | 6 (0.9) |
| No | 637 (99.1) |
| **Feeling of green spaces improving in neighbourhood/community (n = 1914)** | |
| Yes | 12 (0.6) |
| No | 1902 (99.4) |

green spaces were depleting in their neighbourhood/community had 0.3 times (OR = 0.32; 95% CI: 0.25, 0.41) lower odds of developing green spaces at home compared to household heads who felt differently. Household heads who were worried about the depletion, however, had 1.9 times (OR = 1.94; 1.23, 3.05) higher odds of developing green spaces at home compared to those who were not worried about the depletion. Household heads who felt green spaces were not improving in their neighbourhood/community had 12.5 times (OR = 12.48; 95% CI: 3.11, 50.05) higher odds of developing green spaces at home compared to household heads who felt differently. In neighbourhoods with moderate and high level of greenness, household heads had respectively 0.5 times (OR = 0.54, 95% CI: 0.38, 0.77) and 0.2 times (OR = 0.22, 95% CI: 0.16, 0.31) lower odds of developing green spaces at home compared to household heads in neighbourhoods with low levels of greenness.

Figs 2 to 4 depicts the level of greenness in GAMA for the years 2000, 2015 and 2020. These figures suggest a massive depletion and degradation of Accra's green spaces over the last two decades. The level of greenness for GAMA in 2000 is depicted in Fig 2 and shows 47% of

**Table 6. Association of green space use with socio-demographic characteristics of household heads (n = 600).**

| Characteristics | Odds ratio (95% CI) | Adjusted Odds ratio (95% CI) |
|---|---|---|
| **Gender** | | |
| Male | 1.00 | 1.00 |
| Female | 5.91 (2.83, 12.35) | 6.92 (3.13, 15.26) |
| **Age group (years)** | | |
| ≤ 30 | 1.00 | 1.00 |
| 30–49 | 3.08 (1.65, 5.76) | 2.94 (1.48, 5.84) |
| ≥ 50 | 3.05 (1.46, 6.40) | 3.10 (1.36, 7.06) |
| **Educational level** | | |
| No education/Primary | 1.00 | 1.00 |
| JHS | 0.79 (0.24, 2.62) | 0.91 (0.26, 3.15) |
| SHS/Secondary/Technical | 0.51 (0.17, 1.51) | 0.40 (0.13, 1.24) |
| Tertiary | 0.37 (0.12, 1.15) | 0.52 (0.16, 1.71) |
| **Ethinicity** | | |
| Akan | 1.00 | 1.00 |
| Ga/Dangme | 0.1 (0.13, 0.55) | 0.29 (0.14, 0.64) |
| Ewe/Guan | 0.44 (0.19, 1.02) | 0.74 (0.30, 1.84) |
| Mole-Dagbani/Grusi/Gruma/Mande | 1.08 (0.30, 3.96) | 0.99 (0.26, 3.82) |
| Other | 0.23 (0.06, 0.90) | 0.14 (0.03, 0.67) |
| **Religion** | | |
| Christian | 1.00 | 1.00 |
| Muslim | 1.07 (0.44, 2.61) | 0.74 (0.28, 1.93) |
| Traditional/Spiritualist/No religion/Other | 0.10 (0.02, 0.51) | 0.11 (0.02, 0.62) |
| **Marital status** | | |
| Never married | 1.00 | 1.00 |
| Married/ Living together | 1.92 (1.08, 3.42) | 1.88 (1.01, 3.52) |
| Divorced/Separated/ Widowed | 1.88 (0.69, 5.08) | 1.90 (0.66, 5.46) |
| **Occupation** | | |
| Skilled and unskilled manual | 1.00 | 1.00 |
| Professional/Technical/Managerial | 0.21 (0.09, 0.49) | 0.36 (0.15, 0.90) |
| Clerical | 0.63 (0.07, 5.33) | 0.35 (0.04, 3.17) |
| Sales and services | 0.76 (0.35, 1.66) | 0.95 (0.41, 2.17) |
| Unemployed | 0.59 (0.23, 1.53) | 0.72 (0.26, 2.01) |
| Agriculture and Other | 0.30 (0.10, 0.94) | 0.82 (0.23, 2.87) |

Models adjusted for population density of district, income level of district, and neighbourhood greenness.

moderately healthy to very healthy vegetation and 23% of unhealthy vegetation (S2 Table in S1 File). High indices of healthy to very healthy vegetation were noted in the north eastern part of GAMA which encompasses Ayawaso West and a major area of Ayawaso East and North, and La Dadekotopon. Areas with bare/no vegetation occupied 30% of GAMA (S2 Table in S1 File) and were largely found in the south western part of GAMA which encompassed Ayawaso Central, Ablekuma Central, AMA, and part of Korle Klottey, Ayawaso West and La Dadekotopon.

The level of greenness for GAMA in 2015 is depicted in Fig 3 and shows 49% of moderately healthy to very healthy vegetation (S2 Table in S1 File). High indices of healthy to very healthy vegetation were again noted in the northern and eastern part of GAMA with the same affected districts with the exception of Ayawaso North, which lost virtually all the greenness. The map showed an unhealthy vegetation of 26% for 2015 compared to 23% in 2000 (S2 Table in S1

**Table 7. Association of green space development at home and socio-demographic characteristics of household head (n = 1914).**

| Characteristics | Crude Odds ratio (95% CI) | Adjusted Odds ratio (95% CI) |
|---|---|---|
| **Gender** | | |
| Male | 1.00 | 1.00 |
| Female | 1.21 (0.97, 1.52) | 1.24 (0.97, 1.57) |
| **Age group** | | |
| $\leq$ 30 | 1.00 | 1.00 |
| 30–49 | 1.00 (0.74, 1.34) | 0.99 (0.72, 1.37) |
| $\geq$ 50 | 0.68 (0.50, 0.93) | 0.60 (0.43, 0.85) |
| **Educational level** | | |
| No education/Primary | 1.00 | 1.00 |
| JHS | 0.35 (0.22, 0.55) | 0.43 (0.27, 0.69) |
| SHS/Secondary/Technical | 0.27 (0.18, 0.42) | 0.35 (0.22, 0.56) |
| Tertiary | 0.13 (0.07, 0.20) | 0.21 (0.13, 0.34) |
| **Ethinicity** | | |
| Akan | 1.00 | 1.00 |
| Ga/Dangme | 1.22 (0.93, 1.61) | 1.10 (0.81, 1.48) |
| Ewe/Guan | 0.60 (0.44, 0.81) | 0.59 (0.43, 0.82) |
| Mole-Dagbani/Grusi/Gruma/Mande | 1.79 (1.21, 2.65) | 1.11 (0.72, 1.70) |
| Other | 0.79 (0.41, 1.52) | 0.59 (0.29, 1.20) |
| **Religion** | | |
| Christian | 1.00 | 1.00 |
| Muslim | 2.48 (1.69, 3.62) | 1.54 (1.02, 2.32) |
| Traditional/Spiritualist/No religion/Other | 0.71 (0.35, 1.45) | 0.53 (0.25, 1.13) |
| **Marital status** | | |
| Never married | 1.00 | 1.00 |
| Married/ Living together | 1.04 (0.81, 1.34) | 0.90 (0.68, 1.18) |
| Divorced/Separated/ Widowed | 1.03 (0.71, 1.50) | 0.90 (0.60, 1.35) |
| **Occupation** | | |
| Skilled and unskilled manual | 1.00 | 1.00 |
| Professional/Technical/Managerial | 1.05 (0.46, 2.40) | 1.14 (0.49, 2.70) |
| Clerical | 1.59 (1.12, 2.27) | 1.17 (0.80, 1.72) |
| Sales and services | 1.84 (1.26, 2.68) | 1.28 (0.85, 1.91) |
| Unemployed | 1.33 (0.85, 2.08) | 1.10 (0.68, 1.78) |
| Agriculture and Other | 0.71 (0.42, 1.22) | 0.59 (0.33, 1.07) |

Models adjusted for availability of green spaces in neighbourhood, income level of district, and neighbourhood greenness.

File). Areas with bare/no vegetation which occupied 25% of GAMA but with a visible increase in the pixel count (S2 Table in S1 File) were again largely located in the south western part with the same affected districts together with Ablekuma North and Okaikwei North. The lower percentage recorded in 2015 could, however, be due to the satellite technology used i.e. Landsat 8 as against Landsat 7 in 2000.

The level of greenness for GAMA in 2020 is shown in Fig 4 presented and shows 18% of moderately healthy to very healthy vegetation (S2 Table in S1 File). With the exception of some part of Ayawaso West district, virtually all the healthy to very health vegetation recorded in the north eastern part of GAMA during 2000 and 2015 were lost. The map showed an unhealthy vegetation of 43% compared to 26% in 2015 and 23% in 2000 (S2 Table in S1 File).

**Table 8. Association of green space development at home, and characteristics of district/neighbourhood and changing patterns of green spaces in neighbourhood/community.**

| Characteristics | Crude Odds ratio (95% CI) | Adjusted Odds ratio (95% CI) |
|---|---|---|
| **Population density of district** | | |
| Medium (1000–5000) | 1.00 | 1.00 |
| High (5001–10000) | 1.12 (0.81, 1.55) | 0.62 (0.37, 1.05) |
| Very high (10001–20000) | 1.83 (1.36, 2.46) | 1.03 (0.66, 1.60) |
| Extremely high (> 20000) | - | |
| **Income level of district** | | |
| Low-income | 1.00 | 1.00 |
| Middle-income | 0.42 (0.30, 0.59) | 1.26 (0.85, 1.86) |
| High-income | 0.27 (0.18, 0.41) | 0.97 (0.63, 1.50) |
| **Neighbourhood greenness** | | |
| Low | 1.00 | 1.00 |
| Moderate | 0.40 (0.29, 0.57) | 0.54 (0.38, 0.77) |
| High | 0.16 (0.11, 0.22) | 0.22 (0.16, 0.31) |
| **Feeling of green spaces depleting in neighbourhood/ community** | | |
| No | 1.00 | 1.00 |
| Yes | 0.43 (0.34, 0.54) | 0.32 (0.25, 0.41) |
| **Worried about the depletion** | | |
| No | 1.00 | 1.00 |
| Yes | 1.38 (0.99, 1.93) | 1.94 (1.23, 3.05) |
| **Local authority or community leaders taking steps to stem depletion** | | |
| Yes | 1.00 | 1.00 |
| No | 1.11 (0.20, 6.10) | - |
| **Feeling of green spaces improving in neighbourhood/ community** | | |
| Yes | 1.00 | 1.00 |
| No | 11.37 (3.06, 42.20) | 12.48 (3.11, 50.05) |

Models adjusted for availability of green spaces in neighbourhood, income level of district, and neighbourhood greenness.

Areas with bare/no vegetation which occupied 39% of GAMA (S2 Table in S1 File) were largely in the south western part.

Between 2000 and 2015, Ablekuma North and Okaikwei North lost virtually all its moderately healthy to very healthy vegetation. Ablekuma West maintained its moderately healthy to very healthy vegetation during 2000 and 2015. Korle Klottey district maintained its healthy to very healthy vegetation in the northern part of the district during 2000 and 2015. The greenness in all these districts appeared lost by 2020.

## Discussion

### Validity issues

The sampling strategy of the survey together with the high response rate (95.7%) achieved minimizes selection bias. In each selected household visited, we interviewed the household head and another adult member of the household where available independently to ensure the responses of one member is not influenced by the other. The interviewers were also trained by

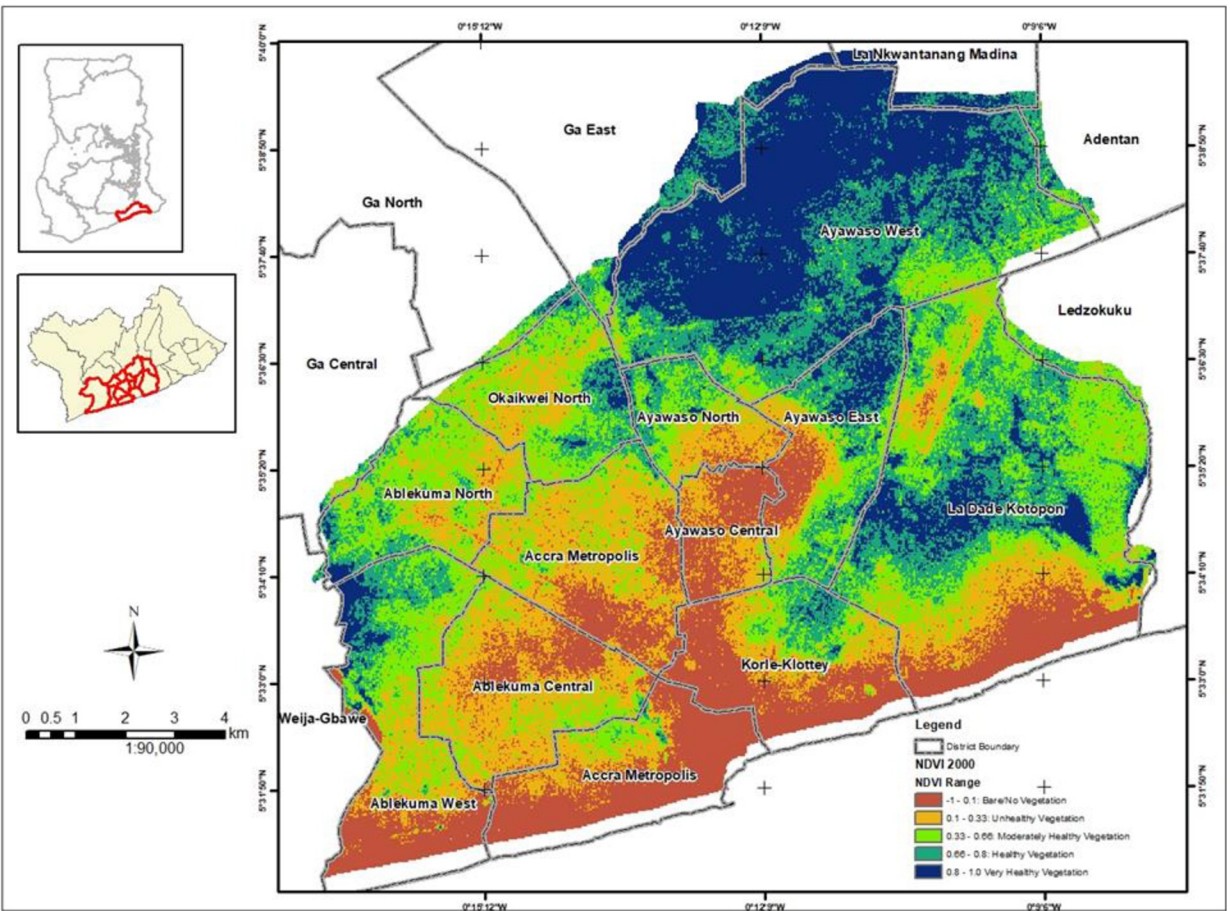

**Fig 2. Neighbourhood greenness in GAMA for the year 2000.**

the study investigators extensively and guarantees the collection of reliable information from study participants. Information bias is thus minimized in the study. We adjusted for potential confounding variables including district level contextual factors i.e. population density, income level and neigbourhood greenness in the in the multivariable analysis. We used principal component analysis (PCA) to create an index of the state of mental health from use of green spaces. PCA allows maximization of the information we keep without using variables that will cause multicolinearity and without having to choose one variable among many. We generated NDVI maps for the years 2000, 2015 and 2020 which measures greenness and density of the vegetation in GAMA to enable us link the extent of greenness in GAMA with residents' perceptions on the availability and state of green spaces in the area, and changing patterns of green spaces in the neighbourhood/community qualitatively.

## Synthesis with previous knowledge

We found green spaces in Accra to be depleting, in poor condition (i.e. lost their greenness) and with city authorities not taking any concrete steps to either stem the depletion or restore the few remaining into good condition. Figs 2 to 4 is a testament to the green space depletion over the last two decades in GAMA. This finding is consistent with studies conducted in some cities of South Africa; Lagos, Nigeria; and Kumasi, Ghana that found green spaces in these cities to be declining [26–28]. According to Amoako & Korboe [26], in Kumasi, Ghana, hailed as

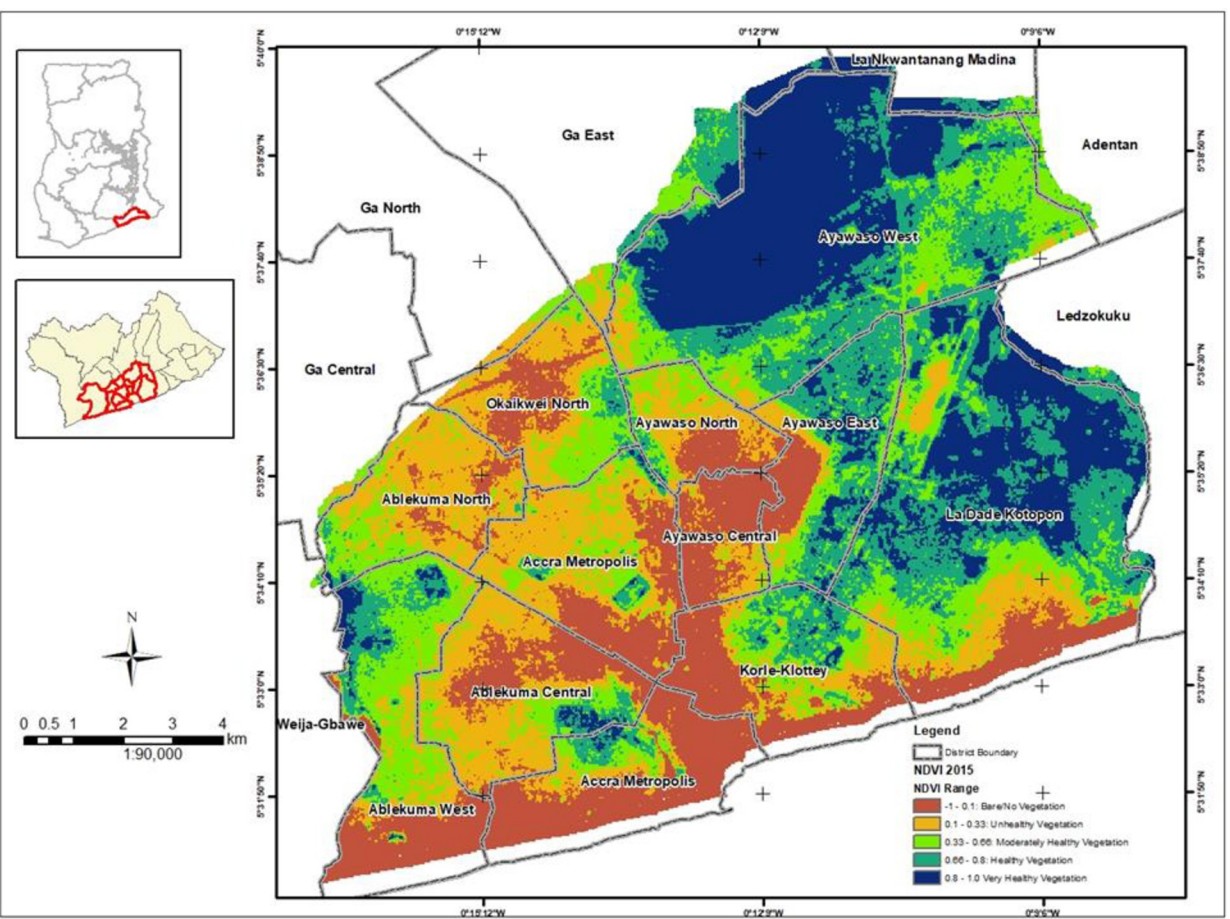

**Fig 3. Neighbourhood greenness in GAMA for the year 2015.**

the Garden City, has seen its green spaces decline to around 10.7% of the total land area of the city. Green spaces depletion in Accra has been attributed to rapid population growth and the growing built-up areas which occurs at the expense of vegetation and grassland destruction [29]. Lack of awareness about the value and importance of green spaces is also a reason for its clearance for residential and commercial buildings. Surprisingly, we found residents who felt green spaces were depleting in their community to be less likely to develop green spaces at home to mitigate the problem. In a study conducted in Kumasi, Ghana [30], the authors found residents demonstrating a low appreciation of urban green spaces and would rather use land to meet their basic needs than develop them into green spaces. According to Cobbinah [31], this situation is very common in areas where locals perceive urban planning as an elitist endeavor. However, residents who were worried about the depletion of green spaces in their community were found to be more likely to develop green spaces at home. Also, residents who felt the local authority or community leaders were not taking any steps to stem depletion of green spaces in their community were more likely to develop green spaces at home. Additionally, residents who felt green spaces were not improving in their community were more likely to develop green spaces at home. It is very likely that, this group of residents see green spaces as more than a recreational amenity and were possibly aware of the physical and psychological health benefits of green spaces and hence, the willingness to develop green spaces at home for improved health.

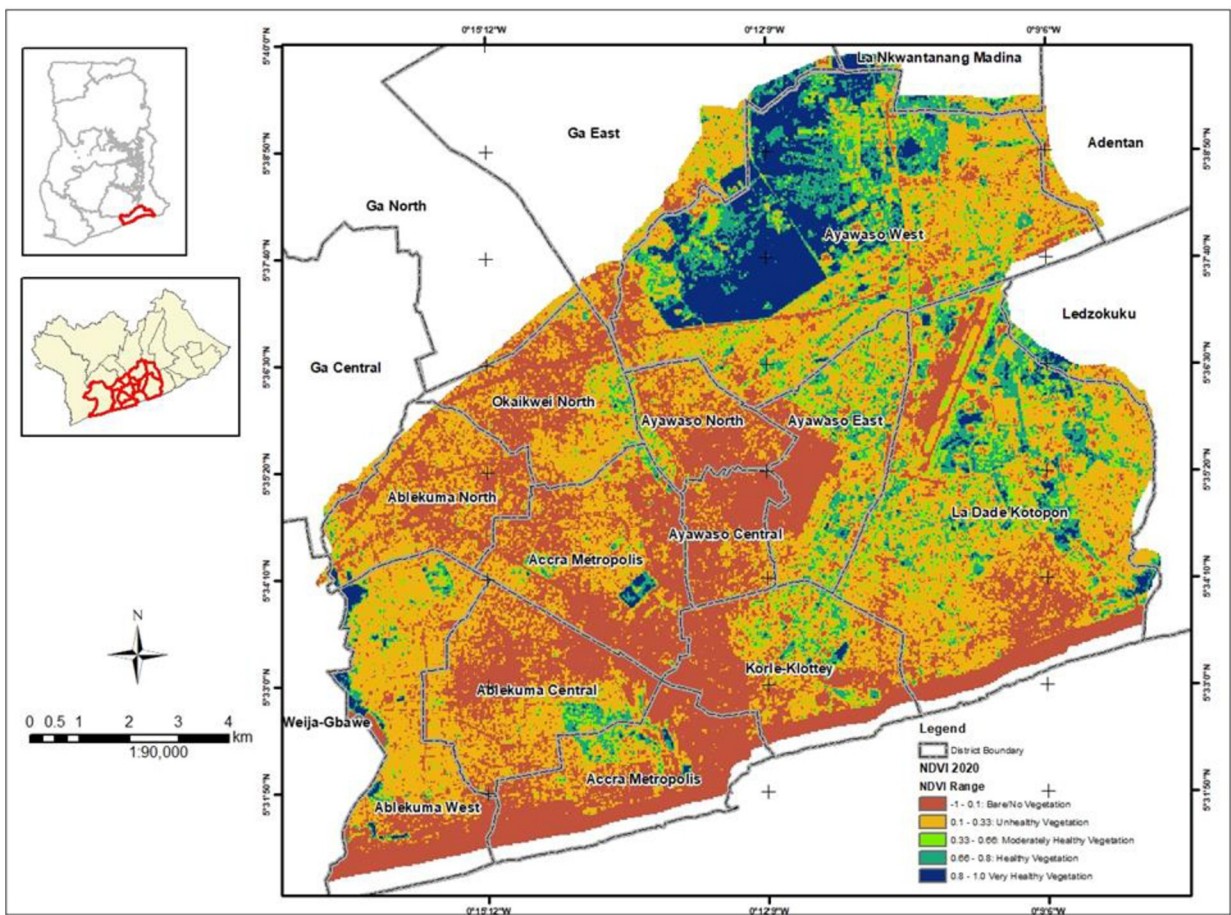

**Fig 4. Neighbourhood greenness in GAMA for the year 2020.**

Urban green spaces have great potential for promoting physical activity with cardiovascular health benefits. We found use of green spaces to be very low in Accra and could be as a result of lack of awareness of the benefits of green spaces among Accra residents or the poor state of these green spaces. Some authors [32,33] have suggested that increasing awareness of the benefits of green spaces may increase people's willingness to use them. Also, according to a study conducted in Addis Ababa, Ethiopia [34], individuals who have positive perceptions about the health benefits of green spaces were more than two times likely to visit green spaces compared to individuals with negative perceptions.

A very small proportion (5%) of green space users in Accra reported witnessing improvement in their physical health from use of green spaces. Our finding is in contrast to the findings of a study conducted in Perth, Western Australia [35] that observed a significant positive relationship between green space use and overall health. In this study, respondents who considered nearby green space to be usable were twice as likely to report better overall health than those who did not. Almost half (47%) of green space users in Accra reported witnessing a very high improvement in their mental health from use of green spaces. This finding is consistent with the findings of several studies [36–38] that found use of neighbourhood green spaces to exert a strong effect on self-reported mental health than physical health. Several epidemiological studies have also reported positive relationship between availability of green spaces in residential neighbourhoods and mental health [39–42]. The pathways for improved mental

health from use of green spaces are deriving satisfaction, finding happiness, developing social connections, guarding against loneliness, feeling of a sense of belonging, and easing stress from use of green spaces. Half or more of our study respondents who reported using green spaces identify with these benefits. Kaźmierczak [43] reported that, local parks in inner-city neighbourhoods provides an important opportunity for developing new social ties and strengthening of existing ones.

Of the proportion of study participants who reported developing green spaces in their home, over half of them identified improved air quality as the main reason for developing green spaces at homes. Green spaces have been found to mitigate environmental hazards with air quality improvement through changes in pollutant concentrations considered as one of the ecosystem services of green spaces [44].

Female residents of Accra were more likely to use green spaces and to develop green spaces at home than male residents. Studies have shown that, men and women perceive and use green space differently. Cohen et al. [45] found women to be significantly underrepresented in urban parks compared to men and that they were less likely to engage in vigorous physical activity in these parks. This observation is, however, contrary to our findings. A study conducted in Denmark [46], however, found women ranking all predefined activities in urban green spaces as more important compared to men and as a result were more likely than men to utilize and enjoy urban green spaces.

Adults and middle age residents of Accra were more likely to use green spaces compared to adolescents and young adult (30 years and below). Adults and middle age people in Accra are likely to use green spaces because they may be aware of the benefits of green space use for their health at that age compared to adolescents and young adults. The benefits envisaged by them may include increase physical activity levels, improve psychological well-being, and reduce risk of cognitive decline. According to Van den Berg et al. [47], passive contact with green spaces affects the psychological restorative system, and lowers blood pressure and stress levels. Hartig et al. [48] also reported that connecting with nature can improve functional and emotional health, lower blood pressure and improve cognition. According to Coon et al. [49], engaging in physical activity in a green space increases feelings of re-vitalization while also reducing tension, confusion, anger and depression. A large cohort study of middle-aged women in the United States [50] that investigated the association of green space exposure with cognitive function found increased exposure to residential green space to be associated with modest benefits in cognition function. However, surprisingly, we found middle age household heads (50 years and above) to be less likely to develop green spaces at home compared to their counterparts who were young adults with the young adults likely motivated by the aesthetics properties of green spaces.

Married or cohabiting residents were more likely to use green spaces compared to residents who have never married. Our finding is consistent with the findings of a study conducted in Kisumu and Eldoret, Kenya [51] that found respondents who were divorced, separated and widowed to be 4.71 times more likely to visit green spaces weekly than unmarried respondents (AOR = 4.71, 95% CI: 1.276, 17.356). Compared to skilled and unskilled manual workers, we found professional, technical and managerial workers to be less likely to use green spaces and could be attributed to white collar workers having limited time to access green spaces due either their busy work schedules or engagement in other leisure activities such as playing tennis.

Compared to uneducated or primary level educated household heads, junior and senior high school, secondary/technical and tertiary educated household heads were less likely to develop green spaces at home. The willingness of persons of lower education to develop green spaces at home could be driven by aesthetic purposes. A study conducted in Denmark

Schipperijn et al. [46] reported that individuals with lower levels of education visited urban green spaces less frequently than those with higher levels of education. Ewe/Guan household heads were less likely to develop green spaces at home compared to Akan household heads. In Ghana, Akans are known to have the tendency to build plush houses with green spaces to showcase their wealth and could account for the finding of this study. Ethnic differences in green space availability and use have been reported by some studies [52–54] and could also explain the findings of this study.

Muslim household heads were more likely to develop green spaces at home compared to Christian household heads. Whereas Christians have no restrictions in their interactions in public spaces, Muslims, have some level of restrictions and could explain why they were more likely to develop green spaces at home. Also, most Christians frequently visit public green spaces such as parks and gardens for prayers and spiritual meditation as observed by a study conducted in Bulawayo, Zimbabwe [55] and could explain why they were less likely to develop green spaces at home. A study conducted in Leicester, UK noted religious differences in green space distribution, access and use [53] and could also explain the findings of this study.

In neighbourhoods with moderate and high level of greenness, household heads were less likely to develop green spaces at home compared to their counterparts in neighbourhoods with low levels of greenness. This finding is to be expected as it means residents in areas with high and moderate levels of greenness have greenness all around their surroundings to access and use for the needed benefits and hence, there is no reason to develop green spaces except for beautification of the house.

## Conclusion

Socio-demographic characteristics such as age, educational level, ethnicity, religion and marital status were found to influence use and development of green spaces in the Accra population. Population density, income level and level of greenness of districts in Accra were also important determinants of green space development at home. Accra witnessed massive depletion and degradation of its green spaces between the period 2000 to 2020. The findings of the survey can inform policy action for mitigating the depletion and degradation of green spaces in African cities as well as promoting the use and development of green spaces for public health gains. The study also provides preliminary data for conducting a large-scale epidemiological study to investigate the influence of urban green spaces on cardiovascular and mental health an African population.

## Supporting information

**S1 File. Sampling strategy and NDVI classification.**
(DOCX)

## Author Contributions

**Conceptualization:** A. Kofi Amegah, Samuel Agyei-Mensah.

**Data curation:** A. Kofi Amegah, Victor Owusu, Lucy Afriyie, Elvis Kyere-Gyeabour, Samuel Agyei-Mensah.

**Formal analysis:** A. Kofi Amegah.

**Funding acquisition:** A. Kofi Amegah, Samuel Agyei-Mensah.

**Investigation:** A. Kofi Amegah, Patrick Osei-Kufuor, Samuel Agyei-Mensah.

**Methodology:** A. Kofi Amegah, Lucy Afriyie, Elvis Kyere-Gyeabour, Patrick Osei-Kufuor, Samuel Agyei-Mensah.

**Project administration:** A. Kofi Amegah, Samuel Agyei-Mensah.

**Resources:** A. Kofi Amegah.

**Supervision:** A. Kofi Amegah, Victor Owusu.

**Validation:** A. Kofi Amegah.

**Visualization:** A. Kofi Amegah.

**Writing – original draft:** A. Kofi Amegah, Kelvin Yeboah.

**Writing – review & editing:** A. Kofi Amegah, Kelvin Yeboah, Victor Owusu, Lucy Afriyie, Elvis Kyere-Gyeabour, Desmond C. Appiah, Patrick Osei-Kufuor, Samuel K. Annim, Samuel Agyei-Mensah, Pierpaolo Mudu.

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
