## [Decision Letter · Decision Letter 0]

25 Jan 2023

PONE-D-22-22097Socio-demographic and neighbourhood factors influencing urban green space use and development at home: A population-based survey in Accra, GhanaPLOS ONE

Dear Dr. Amegah,

Thank you for submitting your manuscript to PLOS ONE. After careful consideration, we feel that it has merit but does not fully meet PLOS ONE’s publication criteria as it currently stands. Therefore, we invite you to submit a revised version of the manuscript that addresses the points raised during the review process.

We look forward to receiving your revised manuscript.

Kind regards,

Jing Cheng

Academic Editor

PLOS ONE

Journal Requirements:

2. Please provide additional details regarding participant consent. In the ethics statement in the Methods and online submission information, please ensure that you have specified whether: 1) whether the ethics committee approved the verbal/oral consent procedure, 2) why written consent could not be obtained, and 3) how verbal/oral consent was recorded. If your study included minors, please state whether you obtained consent from parents or guardians in these cases. If the need for consent was waived by the ethics committee, please include this information.

4. We note that Figures 1-4 in your submission contain [map/satellite] images which may be copyrighted. All PLOS content is published under the Creative Commons Attribution License (CC BY 4.0), which means that the manuscript, images, and Supporting Information files will be freely available online, and any third party is permitted to access, download, copy, distribute, and use these materials in any way, even commercially, with proper attribution. For these reasons, we cannot publish previously copyrighted maps or satellite images created using proprietary data, such as Google software (Google Maps, Street View, and Earth). For more information, see our copyright guidelines: http://journals.plos.org/plosone/s/licenses-and-copyright.

 a. You may seek permission from the original copyright holder of Figures 1-4 to publish the content specifically under the CC BY 4.0 license. 

Reviewers' comments:

Reviewer's Responses to Questions

**Comments to the Author**

1. Is the manuscript technically sound, and do the data support the conclusions?

Reviewer #1: Yes

Reviewer #2: Yes

2. Has the statistical analysis been performed appropriately and rigorously? 

Reviewer #1: Yes

Reviewer #2: Yes

3. Have the authors made all data underlying the findings in their manuscript fully available?

Reviewer #1: Yes

Reviewer #2: No

4. Is the manuscript presented in an intelligible fashion and written in standard English?

Reviewer #1: Yes

Reviewer #2: Yes

5. Review Comments to the Author

Reviewer #1: The paper focuses on an important topic and the findings could contribute to promoting the establishment, maintenance and use of green spaces in Ghana and especially in the capital city of Ghana (Accra). The authours may wish to address the following issues:

1. Abstract: It may be useful to indicate the statistical analysis employed in the study

2. Study Design and Methods: It might be useful to explain why the researchers interviewed an additional adult member in in households where there were other adults members and who were willing to participate in the study. Also, what procedure was used in selecting the additional adult member for interview?

3. It will be informative to explain how the spacial data on green spaces was analyzed

4. Results: Table 2. The authours might wish to explain why they separated the household heads from the entire study participants into a different column "Household Heads".

*It may also be informative for the authours to do cross tabulation of the socio-demographic characteristics against the exposure of interest to determine those that are significantly associated with the exposure of interest rather than the use of only frequencies and percentages. This should inform which of these variable are included in the logistic regression analysis. *Given the multiple sub-level tabulations that have been done, and the small numbers of responses to such questions (n), the use of proportions in the interpretation of the results is misleading.

Table 3. The responses to the question on "use of green spaces in community/neighborhood" is confusing. The authours may wish to revise and present a more clearer table

5. Conclusion: The conclusion should state clearly which socio-demographic factors influence use and development of green spaces in Accra

6. The manuscript will benefit substantially from professional proof reading

Reviewer #2: PLOS ONE

PONE-D-22-22097

Socio-demographic and neighbourhood factors influencing urban green space use and

development at home: A population-based survey in Accra, Ghana

The study combines individual level primary data with contextual secondary data to examine factors associated with access and use of green spaces in Accra, the capital city of Ghana. This is a timely and important study given its relevance to awareness of the fast depleting green spaces in urban environments. However, there are some critical issues that need addressing.

Abstract, provide a brief background of the study area (Accra) to help the reader appreciate the context of the study.

Introduction – Paragraph 2, the authors should use statistical data from the literature to evidence the decline in urban green spaces.

Line 73, “…, it often does NOT translate …” Not is missing.

Line 87 – 88, the authors claim that two studies have been done in Accra but they are narrow in scope. The authors should explain why those studies were narrow in scope, how their study will help fill those gaps.

Introduction, the authors need to provide a convincing justification for the choice of Accra for such a study. Note that the study could have been done in any of the cities in the country.

Lines 107 – 111, the authors should provide a scientific justification with literature evidence to support the classification of population density and income levels of districts. What does the researchers mean by high-income neighbourhoods and how does it translate into income levels?

Lines 116 – 127, when was the data collected?, what data was collected? What were the background characteristics, e.g., age of the respondent, etc…?

Lines 121-123, given the sampling allocation procedure described in Supplementary Table 1, it is not correct to state that “…10 households were selected from the listed households…”. Rather say, 2000 households were selected, allocated proportionately to the number of households in each district.

Lines 121- 123, Describe the sampling procedure used to select 10 households from each EA, and how that leads to a selection of a representative sample for generalisability.

Lines 123-126, What was the relevance of interviewing two people from the same household since it introduces dependence and violates the assumption of statistical independence? Note that interviewing household members separately does not rule out the problem of statistical dependence, since household members have similar or same exposure to access to green space and other factors.

Lines 123-126, What criteria was used to select the second person from the household?

Line 125, the study conducted is quantitative in nature, however, the authors claim that a semi-structured instrument was used for data collection. How is this possible since semi-structured interview guides are used for qualitative studies?

Line 126, define GAMA before using the abbreviation.

Lines 132 – 141, there is potential high multicollinearity in the indicators and scores described. Summing to generate an overall score, high scores in one indicator has the potential to cancel out the scores for another indicator, for example, scoring 2 + 2 + 2 + 2 = 8, also 3 + 3 + 1 + 1 = 8. In this case, using a method of differential weighting such as factor analysis circumvents this problem, and allocate weights due to variation (variances) in the responses. The authors should explain the robustness and limitations of the approach they adopted.

Lines 144 – 158, the authors should explain what NVDI is, what they were used to proxy in this study, with a justification and how the 2000, 2015 and 2020 data were linked to a study conducted in 2021/22?

Why were some of the contextual factors extracted at the district level and others at the community level?

Ethical consideration

The authors stated that verbal informed consent was obtained. They should indicate how it was documented and witnessed. If the study included minors, state how informed consent was obtained. State the date ethical clearance was obtained, the period of data collection, the clearance number to show proof that clearance was obtained before data collection.

Statistical analysis

• “Multivariable binary logistic regression adjusting for potential confounders…” This indicates that there were primary factors upon which the research focuses. Thus, clarify what the primary factors and confounders are.

• What is (are) the outcome variable(s) of interest? Clearly state and indicate how they were classified into 0 and 1 (binary).

• As indicated earlier, two respondents were sampled from each household. This introduces dependence in the data structure which results in a more complex error structure than in a standard ordinary least squares model. When a standard ordinary least squares model is fitted in such case, it could result in biased standard errors, thus affect the model outcome. Explain how nesting within households was accounted for in this analysis. Could adopt multilevel (mixed effects) modelling approaches.

Results

• Tables 3 should rather focus on the main outcome (dependent) variables used for the multivariable analyses to give clarity. Table 5 should focus on the independent variables, classified into primary factors and confounders. Tables 3, 4, 5,.. do not add much to achieving the objectives of the study but rather complicates the story, particularly with the varying and low sample sizes in some cases. Also, those indicators presented in Tables 3, 4 and 5 but are not used in the main (multivariable) analysis should also be excluded. They provide some detail but do not add to achieving the objectives of the paper.

• Some of the variables, e.g., NDVI, population density, etc… were employed as contextual factors but the level at which these were link to the individual data for analysis are not explained.

• Interpretation of odds ratios

o the authors should be specific of what the odds are. For example, rounding 4.53 to 5 is not appropriate since the difference of the effects could be substantial.

o Odds should be interpret as odds, for example, females have higher odds of 4.53 of using green space when compared to males.

o The authors should explain the difference between odds ratios, adjusted odds ratios and crude odds ratios and use them consistently.

• Tables 6, 7 and 8 the authors should use stars (*) to indicate the level of significance to provide clarity on which categories significantly different from others.

• Was the analysis presented in Tables 7 and 8 conducted separately? If yes, then that is appropriate. The socio-demographic factors and neighbourhood factors do not influence green space development independently. For example, persons with higher education maybe more likely to reside in high-income and greener neighbourhoods thus influencing their development of green space. As indicated earlier, the neighbourhood (contextual) factors should be linked to the individual data and the analysis conducted and presented in one table.

• Also, account for the contextual factors in the analysis for use of green space.

• Why are the adjusted odds ratios for population density, income level of district and neighbourhood greenness not reported. Were they statistically insignificant? Please, clarify.

• Lines 388-428, the discussions are missed placed. You do not conduct the analysis before you discuss or describe the variables. You discuss the variables in the data section before you conduct the analysis using the variables.

• Validity issues – these should be discussed in the data section not discussion of the results.

Discussion – Synthesis with previous knowledge

Some sections of the discussion re-present the findings. The focus of the discussions should be on the synthesis and new knowledge.

Lines 545-548, the attribution made by the authors could not be correct. In the context of Ghana, persons of higher education may be engaged in activities that limiting the time to access green space but not that they are not aware of the benefits of green space as claimed by the authors. In this regard, it is important that the authors use literature evidence to attribute their findings

The authors do not use any literature evidence to support or dispute the discussion in the last three paragraphs but just present the results. It is strongly recommended that the discussions focuses on the main findings and literature evidence are used to support the findings. In this regard, the discussions could be shortened.

6. PLOS authors have the option to publish the peer review history of their article (what does this mean?). If published, this will include your full peer review and any attached files.

Reviewer #1: **Yes: **Fabian Sebastian Achana

Reviewer #2: No

---

## [Author Response · Author response to Decision Letter 0]

14 Mar 2023

Journal Requirement

Please provide additional details regarding participant consent. In the ethics statement in the Methods and online submission information, please ensure that you have specified whether: 1) whether the ethics committee approved the verbal/oral consent procedure, 2) why written consent could not be obtained, and 3) how verbal/oral consent was recorded. If your study included minors, please state whether you obtained consent from parents or guardians in these cases. If the need for consent was waived by the ethics committee, please include this information.

We have clarified the informed consent issue in the report under the “Ethical consideration” section of the Methods in response to the same concerns raised by reviewer #2. What we meant by verbal informed consent is that information on the informed consent form was read to the participants by the research assistants with those consenting to participate in the study made to sign or thumbprint the form. We have revised the statement to reflect this detail and provided the ethical clearance reference number as well as the date approval was given under the “Ethical Considerations” section of the Methods. The study did not include minors as the inclusion criteria was household heads and any other adult member of the household. As indicated previously the data collection took place in January, 2022.

We have provided the WHO Unit Reference and Purchasing Order in the Funding Information. 

4. We note that Figures 1-4 in your submission contain [map/satellite] images which may be copyrighted. All PLOS content is published under the Creative Commons Attribution License (CC BY 4.0), which means that the manuscript, images, and Supporting Information files will be freely available online, and any third party is permitted to access, download, copy, distribute, and use these materials in any way, even commercially, with proper attribution. For these reasons, we cannot publish previously copyrighted maps or satellite images created using proprietary data, such as Google software (Google Maps, Street View, and Earth). For more information, see our copyright guidelines: http://journals.plos.org/plosone/s/licenses-and-copyright.

The maps in figures 1-4 contain no satellite images from a third party source. Figure 1 is the administrative map of Greater Accra Metropolitan Area produced from shape files digitized by a contributing author of the paper (Elvis Kyere-Gyeabour). Figures 2-4 contain an overlay of the administrative boundaries and computed NDVI of sentinel bands Red and NIR. These maps contain outputs solely produced by the contributing author and have no third party satellite images. All the images contained in figures 1-4 were produced by the contributing author using ArcGIS Desktop version 10.8. Proprietary data such as Google software (Google Maps, Street View, and Earth) were not used in producing these images. There are therefore no copyright issues bordering on the images supplied in figures 1 – 4 and can therefore be published under the Creative Commons Attribution License (CC BY 4.0). We therefore do not need permission from any copyright holder to publish these figures under the CC BY 4.0 license as none of the maps have been copyrighted.

Reviewer #1: 

1. Abstract: It may be useful to indicate the statistical analysis employed in the study

Done

2. Study Design and Methods: It might be useful to explain why the researchers interviewed an additional adult member in households where there were other adults members and who were willing to participate in the study. Also, what procedure was used in selecting the additional adult member for interview?

Reviewr #2 raised similar concerns by pointing to the fact that interviewing two people from the same household introduces dependence and violates the assumption of statistical independence. We do agree with the concerns raised by you both and have subsequently restricted the analysis to only the household heads.

3. It will be informative to explain how the spacial data on green spaces was analyzed

They were analyzed by generating Normalized Difference Vegetation Index (NDVI) maps. This information is captured under “Generation of Normalized Difference Vegetation Index (NDVI) maps” in the Methods section including the formula used for computing the NDVI.

4. Results: Table 2. The authours might wish to explain why they separated the household heads from the entire study participants into a different column "Household Heads".

We have now restricted the analysis to only household heads based on your concerns in (2) and that of Reviewer #2

It may also be informative for the authours to do cross tabulation of the socio-demographic characteristics against the exposure of interest to determine those that are significantly associated with the exposure of interest rather than the use of only frequencies and percentages. This should inform which of these variable are included in the logistic regression analysis. *Given the multiple sub-level tabulations that have been done, and the small numbers of responses to such questions (n), the use of proportions in the interpretation of the results is misleading.

It is important to show the proportions of each characteristic in the population (univariate analysis) as is common practice in scientific research before focusing on the bivariate analysis. The suggested cross-tabulations will generate additional tables that will make the manuscript clumsy.

Table 3. The responses to the question on "use of green spaces in community/neighborhood" is confusing. The authors may wish to revise and present a more clearer table

Done. We have separated the frequency of use and labelled it as such.

5. Conclusion: The conclusion should state clearly which socio-demographic factors influence use and development of green spaces in Accra

Done

6. The manuscript will benefit substantially from professional proof reading

We have proof read the manuscript severally and corrected all the grammatical and syntax errors

Reviewer #2: 

Abstract, provide a brief background of the study area (Accra) to help the reader appreciate the context of the study.

Done

Introduction – Paragraph 2, the authors should use statistical data from the literature to evidence the decline in urban green spaces.

Done

Line 73, “…, it often does NOT translate …” Not is missing.

Done

Line 87 – 88, the authors claim that two studies have been done in Accra but they are narrow in scope. The authors should explain why those studies were narrow in scope, how their study will help fill those gaps.

Ofori et al. paper is misplaced as it was focused on effect of urban green spaces on local small mammal diversity in Accra and hence we have deleted it. We have identified the gaps in Cobbinah et al. 2021 paper and showed how our study fills those gaps.

Introduction, the authors need to provide a convincing justification for the choice of Accra for such a study. Note that the study could have been done in any of the cities in the country.

Done. Please, see last paragraph of Introduction

Lines 107 – 111, the authors should provide a scientific justification with literature evidence to support the classification of population density and income levels of districts. What does the researchers mean by high-income neighbourhoods and how does it translate into income levels?

Done. Please, see the last paragraph of the “Study design and location” section of the Methods

Lines 116 – 127, when was the data collected?, what data was collected? What were the background characteristics, e.g., age of the respondent, etc…?

Done. Please see last paragraph of “Study Population, Sampling and Data Collection” section under Methods. Background characteristics of the respondents are reported in Table 2

Lines 121-123, given the sampling allocation procedure described in Supplementary Table 1, it is not correct to state that “…10 households were selected from the listed households…”. Rather say, 2000 households were selected, allocated proportionately to the number of households in each district.

Done. 

Lines 121- 123, Describe the sampling procedure used to select 10 households from each EA, and how that leads to a selection of a representative sample for generalisability.

Done. It was systematic sampling and has been incorporated in the sentence you recommended in your preceding comment. 

Lines 123-126, What was the relevance of interviewing two people from the same household since it introduces dependence and violates the assumption of statistical independence? Note that interviewing household members separately does not rule out the problem of statistical dependence, since household members have similar or same exposure to access to green space and other factors.

We agree with the concern raised and have subsequently restricted the analysis to only household heads. Please see revised tables 6, 7 and 8

Lines 123-126, What criteria was used to select the second person from the household?

It was based on who was available in the household at the time of our visit. But we agree that interviewing two people from the same household doesn’t enable statistical independence as pointed to us and have subsequently restricted the analysis to only household heads as indicated.

Line 125, the study conducted is quantitative in nature, however, the authors claim that a semi-structured instrument was used for data collection. How is this possible since semi-structured interview guides are used for qualitative studies?

We have revised the text to read “structured questionnaire”

Line 126, define GAMA before using the abbreviation.

Done

Lines 132 – 141, there is potential high multicollinearity in the indicators and scores described. Summing to generate an overall score, high scores in one indicator has the potential to cancel out the scores for another indicator, for example, scoring 2 + 2 + 2 + 2 = 8, also 3 + 3 + 1 + 1 = 8. In this case, using a method of differential weighting such as factor analysis circumvents this problem, and allocate weights due to variation (variances) in the responses. The authors should explain the robustness and limitations of the approach they adopted.

We agree with the reviewer and have subsequently used principal component analysis to generate the index of state of mental health from use of green spaces from the four variables. We have revise the text under the “Assessment of improvements in mental health” section of the Methods accordingly and have also made a comment on this analysis in the “Validity issues” of the Discussion section

Lines 144 – 158, the authors should explain what NVDI is, what they were used to proxy in this study, with a justification and how the 2000, 2015 and 2020 data were linked to a study conducted in 2021/22?

Done. The first part is captured under the “Generation of Normalized Difference Vegetation Index (NDVI) maps” section of the Methods with the second part captured under the “Validity issues” section of the Discussion. 

Why were some of the contextual factors extracted at the district level and others at the community level?

The contextual factors considered were population density, income level and neighbourhood greenness and were all estimated at the district level.

Ethical consideration: The authors stated that verbal informed consent was obtained. They should indicate how it was documented and witnessed. If the study included minors, state how informed consent was obtained. State the date ethical clearance was obtained, the period of data collection, the clearance number to show proof that clearance was obtained before data collection.

What we meant by verbal informed consent is that information on the informed consent form was read to the participants by the research assistants with those consenting to participate in the study made to sign or thumbprint the form. We have revised the statement to reflect this detail and provided the ethical clearance reference number as well as the date approval was given under the “Ethical Considerations” section of the Methods. The study did not include minors as the inclusion criteria was household heads and any other adult member of the household. As indicated previously the data collection took place in January, 2022.

Statistical analysis

• “Multivariable binary logistic regression adjusting for potential confounders…” This indicates that there were primary factors upon which the research focuses. Thus, clarify what the primary factors and confounders are.

We are not sure what the reviewer means by primary factors but in the “Statistical analysis” section we indicated the response (green space use and green space development at home) and predictor (socio-demographic, neighbourhood and health factors) variables. Underneath Tables 6-8, we have indicated the confounders adjusted for in the multivariable analysis. 

• What is (are) the outcome variable(s) of interest? Clearly state and indicate how they were classified into 0 and 1 (binary).

The outcome variables are mentioned in the “Statistical analysis” section. In Tables 6-8, the response/characteristics labelled 1.00 under the Odds Ratio column were classified as 0 and served as reference in the analysis as is customary done with the other response/characteristics classified as 1. We don’t think it is necessary to detail these binary categorizations in the Methods as the tables provide all the information. 

• As indicated earlier, two respondents were sampled from each household. This introduces dependence in the data structure which results in a more complex error structure than in a standard ordinary least squares model. When a standard ordinary least squares model is fitted in such case, it could result in biased standard errors, thus affect the model outcome. Explain how nesting within households was accounted for in this analysis. Could adopt multilevel (mixed effects) modelling approaches.

As indicated we have restricted the analysis to only household heads.

Results

• Tables 3 should rather focus on the main outcome (dependent) variables used for the multivariable analyses to give clarity. Table 5 should focus on the independent variables, classified into primary factors and confounders. Tables 3, 4, 5,.. do not add much to achieving the objectives of the study but rather complicates the story, particularly with the varying and low sample sizes in some cases. Also, those indicators presented in Tables 3, 4 and 5 but are not used in the main (multivariable) analysis should also be excluded. They provide some detail but do not add to achieving the objectives of the paper.

We feel it is necessary to retain all the information in the tables as they answer our research questions and help achieve our objectives contrary to the feelings of the reviewer. Our objective as indicated in the first part of the last paragraph of the Introduction is to “Establish availability, accessibility and use of urban green spaces, the perceived health benefits whilst also attempting to establish the factors influencing use and development of green spaces at home” The information the reviewer is asking us to expunged from the tables answers the first part of the objectives and we are requesting that it is retain irrespective of the low sample sizes in some cases. 

• Some of the variables, e.g., NDVI, population density, etc… were employed as contextual factors but the level at which these were link to the individual data for analysis are not explained.

As indicated previously and now reported under the “Validity issues” section of the Discussion it enables us to link the extent of greenness in GAMA with residents’ perceptions on the availability and state of green spaces in the area, and changing patterns of green spaces in the neighbourhood/community qualitatively. We also adjusted for these contextual factors in the multivariable analysis to help remove their influence on the relationship as they are strongly associated with both the exposure (predictor) and outcome (response) variable

• Interpretation of odds ratios

We have corrected interpretation of the odds ratios wherever they occur in the report.

• the authors should be specific of what the odds are. For example, rounding 4.53 to 5 is not appropriate since the difference of the effects could be substantial.

We have taken note of this comment and made the necessary changes in the report.

• Odds should be interpreted as odds, for example, females have higher odds of 4.53 of using green space when compared to males.

Done

• The authors should explain the difference between odds ratios, adjusted odds ratios and crude odds ratios and use them consistently.

All the odds ratios reported and interpreted in the Results sections are adjusted odds ratios which adjust for the potential confounding variables 

• Tables 6, 7 and 8 the authors should use stars (*) to indicate the level of significance to provide clarity on which categories significantly different from others.

The confidence intervals provide information on the precision of the estimate and also whether the range includes the null value (1.00) or not. We therefore feel we should not reduce this important information provided by the CIs to just significant and not significant, and the level of significance (i.e. p<0.05, 0.01, 0.001) which is at the heart of the abuse of p values in the clinical and epidemiological literature. We think we have made a good case for us to avoid use of the stars. 

• Was the analysis presented in Tables 7 and 8 conducted separately? If yes, then that is appropriate. The socio-demographic factors and neighbourhood factors do not influence green space development independently. For example, persons with higher education maybe more likely to reside in high-income and greener neighbourhoods thus influencing their development of green space. As indicated earlier, the neighbourhood (contextual) factors should be linked to the individual data and the analysis conducted and presented in one table.

Yes, the analyses were conducted separately.

• Also, account for the contextual factors in the analysis for use of green space.

We did as indicated beneath Table 6.

• Why are the adjusted odds ratios for population density, income level of district and neighbourhood greenness not reported. Were they statistically insignificant? Please, clarify.

It was an oversight. For each relationship, we have adjusted for the other two contextual factors in the multivariable analysis. Please, see the revised Table 8.

• Lines 388-428, the discussions are missed placed. You do not conduct the analysis before you discuss or describe the variables. You discuss the variables in the data section before you conduct the analysis using the variables.

We have revised the text.

• Validity issues – these should be discussed in the data section not discussion of the results.

Validity issues borders on the strengths and limitations of the study which the lead and corresponding author have titled as such throughout his publishing career. Strength and limitations of a study are documented under the Discussion section of research papers and not the Data section.

Discussion – Synthesis with previous knowledge

• Some sections of the discussion re-present the findings. The focus of the discussions should be on the synthesis and new knowledge.

We have revised the section extensively to remove the wholesale repetition of results. In some places, however, it is worth pointing to the results briefly before synthesizing with the previous literature. 

• Lines 545-548, the attribution made by the authors could not be correct. In the context of Ghana, persons of higher education may be engaged in activities that limiting the time to access green space but not that they are not aware of the benefits of green space as claimed by the authors. In this regard, it is important that the authors use literature evidence to attribute their findings

We have revised the sentence to reflect the reviewer’s suggestion

• The authors do not use any literature evidence to support or dispute the discussion in the last three paragraphs but just present the results. It is strongly recommended that the discussions focuses on the main findings and literature evidence are used to support the findings. In this regard, the discussions could be shortened

We have supported the findings with appropriate references.

---

## [Decision Letter · Decision Letter 1]

15 May 2023

Socio-demographic and neighbourhood factors influencing urban green space use and development at home: A population-based survey in Accra, Ghana

PONE-D-22-22097R1

Dear Dr. Amegah,

We’re pleased to inform you that your manuscript has been judged scientifically suitable for publication and will be formally accepted for publication once it meets all outstanding technical requirements.

Kind regards,

Jing Cheng

Academic Editor

PLOS ONE

Additional Editor Comments (optional):

Reviewers' comments:

Reviewer's Responses to Questions

**Comments to the Author**

1. If the authors have adequately addressed your comments raised in a previous round of review and you feel that this manuscript is now acceptable for publication, you may indicate that here to bypass the “Comments to the Author” section, enter your conflict of interest statement in the “Confidential to Editor” section, and submit your "Accept" recommendation.

Reviewer #3: All comments have been addressed

Reviewer #4: All comments have been addressed

2. Is the manuscript technically sound, and do the data support the conclusions?

Reviewer #3: Yes

Reviewer #4: Yes

3. Has the statistical analysis been performed appropriately and rigorously? 

Reviewer #3: Yes

Reviewer #4: Yes

4. Have the authors made all data underlying the findings in their manuscript fully available?

Reviewer #3: Yes

Reviewer #4: Yes

5. Is the manuscript presented in an intelligible fashion and written in standard English?

Reviewer #3: Yes

Reviewer #4: Yes

6. Review Comments to the Author

Reviewer #3: the work should contain a flowchart explain the steps of the work in the methods section.

the work should contain a flowchart explain the steps of the work in the methods section.

Reviewer #4: I commend the authors for effectively addressing my comment, demonstrating their dedication and responsiveness to feedback.

7. PLOS authors have the option to publish the peer review history of their article (what does this mean?). If published, this will include your full peer review and any attached files.

Reviewer #3: **Yes: **Mohamed A. E. AbdelRahman

Reviewer #4: No

---

## [Editor Report · Acceptance letter]

16 Jun 2023

PONE-D-22-22097R1 

Socio-demographic and neighbourhood factors influencing urban green space use and development at home: A population-based survey in Accra, Ghana 

Dear Dr. Amegah:

I'm pleased to inform you that your manuscript has been deemed suitable for publication in PLOS ONE. Congratulations! Your manuscript is now with our production department. 

Kind regards, 

on behalf of

Dr. Jing Cheng 

Academic Editor

PLOS ONE